# Reference Values for the German Version of the Quality of Life after Brain Injury in Children and Adolescents (QOLIBRI-KID/ADO) from a General Population Sample

**DOI:** 10.3390/jpm14040336

**Published:** 2024-03-23

**Authors:** Leonie Krol, York Hagmayer, Nicole von Steinbuechel, Katrin Cunitz, Anna Buchheim, Inga K. Koerte, Marina Zeldovich

**Affiliations:** 1Department of Psychology, Clinical Psychology, Experimental Psychopathology, and Psychotherapy, University of Marburg, 35037 Marburg, Germany; 2Georg-Elias-Müller Institute for Psychology, Georg-August-University, 37073 Goettingen, Germany; york.hagmayer@bio.uni-goettingen.de; 3Institute of Psychology, University of Innsbruck, 6020 Innsbruck, Austria; nicole.von-steinbuechel@uibk.ac.at (N.v.S.); katrin.cunitz@uibk.ac.at (K.C.); anna.buchheim@uibk.ac.at (A.B.); 4Department of Psychiatry and Psychotherapy, University Medical Center Goettingen, 37075 Goettingen, Germany; 5cBRAIN/Department of Child and Adolescent Psychiatry, Psychosomatics, and Psychotherapy, LMU University Hospital, Ludwig-Maximilian University, 80337 Munich, Germany; inga.koerte@med.uni-muenchen.de; 6Psychiatry Neuroimaging Laboratory, Department of Psychiatry, Brigham and Women’s Hospital, Mass General Brigham, Harvard Medical School, Boston, MA 02115, USA; 7Faculty of Psychotherapy Science, Sigmund Freud University Vienna, Freudplatz 1, 1020 Vienna, Austria

**Keywords:** reference values, general population, patient-reported outcome measure (PROM), health-related quality of life, children and adolescents, traumatic brain injury

## Abstract

Assessment of health-related quality of life (HRQoL) after pediatric traumatic brain injury (TBI) has been limited in children and adolescents due to a lack of disease-specific instruments. To fill this gap, the Quality of Life after Traumatic Brain Injury for Children and Adolescents (QOLIBRI-KID/ADO) Questionnaire was developed for the German-speaking population. Reference values from a comparable general population are essential for comprehending the impact of TBI on health and well-being. This study examines the validity of the German QOLIBRI-KID/ADO in a general pediatric population in Germany and provides reference values for use in clinical practice. Overall, 1997 children and adolescents aged 8–17 years from the general population and 300 from the TBI population participated in this study. The questionnaire was tested for reliability and validity. A measurement invariance (MI) approach was used to assess the comparability of the HRQoL construct between both samples. Reference values were determined by percentile-based stratification according to factors that significantly influenced HRQoL in regression analyses. The QOLIBRI-KID/ADO demonstrated strong psychometric properties. The HRQoL construct was measured largely equivalently in both samples, and reference values could be provided. The QOLIBRI-KID/ADO was considered reliable and valid for assessing HRQoL in a general German-speaking pediatric population, allowing for clinically meaningful comparisons between general and TBI populations.

## 1. Introduction

Pediatric traumatic brain injury (TBI) is a serious and common injury in children and adolescents [1,2]. It can have a variety of short- and long-term consequences for affected individuals and their families, as described in, e.g., [3,4]. The perspective on TBI shifted from being an acute injury to a chronic condition with far-reaching impacts on people’s lives [4]. For instance, children who have suffered a TBI may have impaired cognitive and executive functioning and memory issues [5,6]. They are also at increased risk for developing mental health and behavioral problems at both the subclinical and clinical level [7]. In fact, experiencing TBI during childhood increases the likelihood of psychosocial problems in adulthood [8]. TBI can result in functional impairment [9], decreased mobility [10], and reduced health-related quality of life (HRQoL) in children and adolescents [11,12,13].

HRQoL can be defined as “[a] child’s goals expectations, standards or concerns about their overall health and health-related domains“ [14] (p. 2). Although there is no gold-standard definition, the multidimensionality of the construct is a central component [15]. Usually, it comprises physical, social, and psychosocial (including emotional and cognitive) domains [16].

HRQoL can be measured generically and disease-specifically. Generic instruments capture a broad spectrum of quality of life and health status indicators. In addition, they can be applied to both healthy and diseased individuals, so that results are comparable within these groups. In contrast, disease-specific instruments capture specific problems or symptoms of a disease that a generic measure may neglect. Disease-specific measures are therefore only applicable to individuals with a specific condition and can be used to measure changes after treatment [15,17]. It has been shown that disease-specific generic instruments are preferable for various diseases [15,18] as well as specifically for TBI [19] when it comes to accurate differentiation of HRQoL. Thus far, the generic Pediatric Quality of Life Inventory (PedsQL) [20] has been the primary instrument used to assess pediatric HRQoL. Alternatives such as the Child Health Questionnaire (CHQ) [21], KINDL [22], or 36-Item Short Form Health Questionnaire (SF-36) [23] provide additional options, but likewise map generic HRQoL. Until now, there have been no TBI-specific, self-reported HRQoL measures for children and adolescents [24]. This is a notable omission, as HRQoL following TBI is often impaired compared to normative data [11,12,25] or other health conditions such as cardiac or orthopedic conditions [26], as demonstrated by generic assessment measures.

For this reason, the Quality of Life after Brain Injury in Children and Adolescents (QOLIBRI-KID/ADO) [27] was developed. It is the first disease-specific pediatric patient-reported outcome measure (PROM) for this population. The QOLIBRI-KID/ADO comprises 35 items and was developed through a systematic iterative process that involved focus group interviews, international and national expert interviews, and Delphi panels [28]. It is intended for use by children and adolescents aged 8–17 years. The questionnaire is suitable for longitudinal evaluation due to its theoretical alignment with the adult version [29,30] and adoption of the six-factor structure. The QOLIBRI-KID/ADO questionnaire comprises six scales, including Cognition, Self, Daily Life and Autonomy, Social Relationships, Emotions, and Physical Problems. Overall, the instrument demonstrates good to very good internal consistency for the scale scores, with α values ranging from 0.70 to 0.89. Validity analyses indicate weak to moderate convergent validity (PedsQL, r = 0.47 to r = 0.67) and discriminant validity for anxiety and depression (GAD-7, r = −0.31; PHQ-9, r = −0.36) [27].

Although a substantial overlap between the generic PedsQL and the disease-specific QOLIBRI-KID/ADO was found in terms of total scores (r = 0.67) [27], the latter covers aspects that are particularly important to capture after TBI (e.g., impact of TBI on cognition or independence in daily living). Therefore, some items of the QOLIBRI-KID/ADO directly address TBI. Although both questionnaires seem to address similar physical aspects, the QOLIBRI-KID/ADO assesses detailed problems that children and adolescents may experience after an injury, such as “How much do headaches bother you?” and “How much do other kinds of pain (other than headaches) bother you?”, while the PedsQL is more general, stating “I was in pain.” As a result, the QOLIBRI-KID/ADO is longer than the PedsQL, with 35 items compared to 23, but it is better able to detect finer differences in HRQoL problems after TBI.

Assessing and interpreting HRQoL with a PROM can be challenging for patients and healthcare professionals alike [31]. Only clinical evaluation can determine whether a change in post-injury HRQoL compared to the general population is clinically relevant. Therefore, clinicians, researchers, and patients can benefit from using reference values [32]. The aim of this study is thus to provide these reference values using data from a German-speaking general pediatric population sample.

## 2. Materials and Methods

### 2.1. Study Design

Data for the general pediatric population sample were derived from an online panel-based, self-reported, cross-sectional study. The sample of children and adolescents post TBI used for the MI analyses was recruited cross-sectionally using self-report in a face-to-face interview (online or in person).

### 2.2. Participants

#### 2.2.1. General Population Sample

The general population sample was recruited using the services of two marketing agencies (dynata and respondi; https://www.dynata.com, last accessed on 4 December 2023) and https://www.bilendi.co.uk, last accessed on 4 December 2023). These agencies contacted adults with children between the ages of 8 and 17 during March and April 2022. The parents were provided with information about the data collection procedure, including its purpose and privacy policy. Participation took place only after the adults provided their consent for the collection of sensitive health information from their children. Parents reported sociodemographic information, including whether their child had sustained a TBI or was currently experiencing a life-threatening illness. If either situation applied, the survey was terminated. If not, parents were asked to confirm their child’s presence. If the child was unavailable, the survey could be resumed at a later time. If the child was present, they were invited to take part and were referred to the pediatric questionnaires upon agreement. After finishing the questionnaire, parents were given compensation in the form of vouchers or tokens.

The survey was initiated by 5057 individuals, 2164 of whom completed it. Participants were excluded if they provided contradictory responses (one-sided responses regardless of item polarity), completed the survey in under five minutes, provided inconsistent (e.g., selected a disease while reporting that they were completely healthy), unusable (e.g., cryptic comments), or no disease information (e.g., a comment in the text box without specifying a health condition). For further details, see Figure 1A.

#### 2.2.2. TBI Sample

From April 2017 to January 2022, data for the TBI sample were collected at eleven hospitals in Germany. Study details and data collection were communicated to parents, children, and adolescents, who provided written informed consent. To be eligible for the study, participants had to be between 8 and 17 years of age, diagnosed with TBI (at least 3 months and no more than 10 years after injury), have their TBI severity assessed by the Glasgow Coma Scale (GCS) [33] or clinical description of severity, be an outpatient or beginning inpatient discharge, and have the ability to comprehend and respond to questions.

Epilepsy prior to TBI, spinal cord injury, persistent vegetative state (i.e., minimal consciousness according to the GCS), severe premorbid mental disorder (such as psychosis or autism), terminal disease, or very severe polytrauma (as evaluated by the examiner) led to exclusion from the study. Approximately 5000 eligible families were contacted, and 300 children and adolescents were included in the final study sample. For more details, see Figure 1B.

### 2.3. Ethical Approval

Both studies were conducted in compliance with German laws and regulations as well as the ICH Harmonized Tripartite Guideline for Good Clinical Practice (“ICH GCP”) and the World Medical Association Declaration of Helsinki (“Ethical Principles for Medical Research Involving Human Subjects”). Participants and/or legal guardians obtained informed consent according to the German General Data Protection Regulation (GDPR). The Ethics Committee of the University Medical Center Goettingen approved this study (application no. 19/4/18).

### 2.4. Materials

#### 2.4.1. Sociodemographic and Health-Related Data

This study collected sociodemographic information including age, gender, and (parental) education level. Furthermore, the parents of the children included in the general population sample were asked to provide details concerning their children’s health status. Health status consisted of nine categories, including disorders of the central nervous system, abuse of alcohol and/or psychotropic drugs, active or uncontrolled systemic diseases, psychiatric disorders, severe sensory deficits, use of psychotropic or other medications, intellectual disabilities or other neurobehavioral disorders, pre-/peri-/postnatal problems, as well as other issues. If one category was selected, the child was considered to have at least one chronic condition.

Clinical data on TBI severity, time since TBI, and functional recovery/disability were collected in the TBI sample. The King’s Outcome Scale for Childhood (KOSCHI) [34] was utilized to determine functional recovery/disability at testing time, covering the following categories: intact recovery (5b), good recovery (5a), upper moderate disability (4b), lower moderate disability (4a), upper severe disability (3b), lower severe disability (3a).

#### 2.4.2. Quality of Life after Brain Injury in Children and Adolescents (QOLIBRI-KID/ADO)

The QOLIBRI-KID/ADO is a PROM designed to assess TBI-specific HRQoL in children and adolescents between the ages of 8 and 17. The questionnaire comprises 35 items, which respondents answer using a five-point Likert-type scale (“Not at all” = 1, “Slightly” = 2, “Moderately” = 3, “Quite” = 4, “Very” = 5). It covers the following domains: Cognition, Self, Daily Life and Autonomy, Social Relationships, Emotions, and Physical Problems. The first four scales measure satisfaction (“How satisfied are you...?”), and the last two scales measure feelings of being bothered (“How bothered are you by...?”). To ensure consistent interpretation, the items of these last two scales are inversely recoded. The scale scores and the total score are converted linearly to a 0–100 scale, with higher values corresponding to better HRQoL. For the items to be relevant to the general population, the instructions and items with TBI-specific content were modified. Detailed information on the modified wording can be found in Appendix A—Table A1.

### 2.5. Statistical Analyses

#### 2.5.1. Descriptive Statistics, Reliability, and Differential Item Functioning Analyses

Descriptive statistics were calculated, including the mean, standard deviation, and skewness. Skewness was determined to be symmetric (≤|0.5|), moderate (|0.5| < x ≤ |1|), or high (>|1|) [35]. Internal consistency was evaluated using Cronbach’s α at both the scale and total score levels. Additionally, McDonald’s ω was calculated for the scales and the total score. A Cronbach’s α value between 0.7 and 0.95 was considered good [36], while a McDonald’s ω value above 0.8 was deemed to be good [37]. Corrected item–total correlations (CITCs) were computed, and items with values exceeding 0.4 were considered consistent [38].

Differential item functioning (DIF) using logistic ordinal regression approach combined with item response theory (LORDIF) was conducted to examine the appropriateness of aggregating data from children and adolescents from the general population. A second DIF analysis was performed to gain a better understanding of potential item-level differences between the TBI and general population samples. For this purpose, two LORDIF models were compared for each item: one including only the level of the latent trait (i.e., item-level HRQoL), and another including the level of the latent trait, age category (for aggregation of age, i.e., children vs. adolescents) or sample type (for sample differences, i.e., TBI vs. general population), and the interaction of both variables (HRQoL*age category or HRQoL*sample type). DIF was considered absent if a non-significant difference (*p* > 0.01) was found and the associated effect size (McFadden’s pseudo R^2^) was small (R^2^ < 0.05) [39]. In cases where DIF was not present, responses were treated as independent of age and sample type.

#### 2.5.2. Construct Validity of QOLIBRI-KID/ADO in the General Population Sample

The factorial structure was assessed using confirmatory factor analysis (CFA) with a robust weighted least squares estimator for ordinal data. A six-factor model was examined. To assess model goodness of fit, we utilized the following fit indices (desired values are in parentheses): χ2 value (*p* > 0.05), ratio of χ2 value and degrees of freedom (χ2/df ≤ 2) [40], comparative fit index (CFI ≥ 0.95) [41], Tucker–Lewis index (TLI ≥ 0.95) [41], root mean square error of approximation (close to excellent fit RMSEA < 0.05; moderate fit 0.05 ≤ RMSEA < 0.10) [42,43] with 90% confidence interval (CI_90%_), and standardized root mean square residual (SRMR < 0.08) [41]. Scaled χ2-based fit indices were reported (i.e., CFI, TLI, and RMSEA) to ensure the robustness of the results. All indices were considered simultaneously, as the cut-offs for ordinal data should be interpreted with caution [44].

We conducted multiple linear regressions to assess construct validity and identify potential confounders for later stratification of reference values. The QOLIBRI-KID/ADO was utilized as the outcome measure. Gender, age, and health status were treated as covariates, including all possible second-order interactions (such as gender × age and gender × health status). The total score was tested with a significance level of 5% and the scales were tested with a Bonferroni correction (i.e., 5%/6 = 0.8%).

#### 2.5.3. Measurement Invariance across Samples

We conducted a multi-group CFA for ordinal categorical outcomes according to Wu and Estabrook [45], revised by Svetina, Rutkowski, and Rutkowski [46], to test whether the measured construct of HRQoL was the same in the general population sample and the TBI sample. Due to missing responses in the low categories for two items (Orientation, Accomplishment) in the TBI sample, the response categories “Not at all” and “Slightly” were combined into one in both samples to allow for MI measurement.

Three models were estimated with increasingly more restricted parameters [46]: first, we estimated a baseline model without any restrictions; second, we restricted the model for equal thresholds across samples; and finally, we added the restriction for equal loadings. Differences were tested using the scaled χ2 difference test with between-model cut-offs. As between-model cut-offs ∆CFI (<0.01) [47] and ∆RMSEA (≤0.01) [48] were used. We assumed equivalent models if the χ2 difference test was non-significant (*p* > 0.05) and ∆CFI and ∆RMSEA below the cut-offs.

Further analyses were carried out to compare the measurement of the construct between the two samples in cases where the invariance assumption was violated. We compared thresholds between the general population and the TBI sample for significant models, with minimal differences indicated if they did not exceed 5% [49,50]. Additionally, we compared the item factor loadings across the samples.

#### 2.5.4. Reference Values from the General Population Sample

Percentiles indicating the threshold below which a certain percentage of observations fall were used to determine the reference values. The provided data include percentiles at 2.5%, 5%, 16%, 30%, 40%, 50%, 60%, 70%, 85%, 95%, and 97.5%. Values one standard deviation below the reference mean, which corresponds to the 16th percentile for normally distributed data, were deemed clinically relevant.

#### 2.5.5. Software

Analyses were carried out with R version 4.2.3 [51] using the packages table1 [52] for descriptive statistics, psych [53] for psychometric analyses, lordif [39] for DIF analyses, and lavaan [54] for (multigroup) confirmatory factor analysis (CFA).

## 3. Results

### 3.1. Sociodemographic and Health-Related Data

#### 3.1.1. General Population Sample

This study included 1997 children (52.4%) and adolescents (47.6%) from the general population. The average age of the participants was 12.4 years (SD = 2.85), with a balanced gender ratio (50.4% male). Most children and adolescents attended preparatory high school (29.6%), primary school (27.8%), or secondary school (26.7%). Approximately 12.5% of the children and adolescents had at least one chronic health condition. For details of the demographic information, see Table 1.

Of those with a chronic health condition, the majority had only one (11%), with a maximum of three (0.2%). “Other” (4.9%), “intellectual disabilities or other neurobehavioral disorders” (4.7%) and “psychiatric disorders” (2.2%) were the most commonly reported categories.

#### 3.1.2. TBI Sample

A total of 300 children (50.7%) and adolescents (49.3%) who had experienced TBI were included in the analyses. The majority were males (59.3%) and had suffered a mild TBI (71.7%) 4.51 (SD = 2.78) years prior to study enrollment. Most of them achieved a good level of recovery (89.6%; KOSCHI scores of 5a and 5b). Please refer to Appendix A—Table A2 for further details.

### 3.2. Descriptive Statistics, Reliability, and Differential Item Functioning Analyses

The average QOLIBRI-KID/ADO total score for children and adolescents in the general population sample yielded 73.0 (SD = 13.5), which exhibited symmetry (S = −0.38). All items were moderately skewed to the left with a mean skewness of M = 0.74 (SD = 0.31) and a range of −0.27 to −1.74. The internal consistency of the QOLIBRI-KID/ADO total score for the general population sample was excellent, as demonstrated by Cronbach’s α (0.94) and McDonald’s ω (0.95). Alpha coefficients for the scales ranged from 0.80 to 0.86, and McDonald’s ω coefficients were between 0.83 and 0.90. Excluding none of the items improved the Cronbach’s α of a scale. The CITC value for all items was above 0.4, except for the item Orientation, which was already below 0.4 for the TBI version with the CITC [27]. Table 2 presents detailed psychometric properties.

DIF analyses between children and adolescents in the general population sample yielded statistically significant results (*p* < 0.01) for approximately half of the items. These items include Decision between two, Accomplishment, Daily independence, Getting out and about, Manage at school, Decision making, Support from others, Ability to move, Open up to others, Relationship with friends, Attitudes of others, and Clumsiness. However, none of the items had a McFadden’s R^2^ value greater than 0.05. This suggests that the effects were minimal and can be considered negligible. Thus, analyzing aggregated data was considered appropriate. For further information on the DIF analyses results, refer to Appendix A—Table A3.

DIF analyses revealed significant differences (*p* < 0.01) between the TBI and general population samples in most items, including Talking to others, Remembering, Decision-making, Orientation, Accomplishment, Appearance, Self-esteem, Future, Daily independence, Getting out and about, Manage at school, Social activities, Ability to move, Family relationship, Friendships, Loneliness, Clumsiness, Other injuries, Headaches, Pain, and Seeing/Hearing, as well as Life changes. However, most McFadden’s R^2^ values for the significant items were less than 0.01, indicating that these differences are negligible. Exceptions were found for the items Accomplishment, Appearance, Daily independence, Family relationship, and Other injuries, but even in these cases, McFadden’s R^2^ did not exceed 0.05, suggesting that the differences were again negligible. Overall, the samples can be considered comparable, allowing for direct item comparison. See Appendix A—Table A4 for more detailed information.

### 3.3. Construct Validity of QOLIBRI-KID/ADO in the General Population Sample

Results of the CFA were satisfactory for the six-factor structure: χ2(545) = 4500.654, *p* < 0.001, χ2/df = 8.258, CFI = 0.929, TLI = 0.922, RMSEA [90% CI] = 0.060[0.059; 0.062]; SRMR = 0.051. All values met the required cut-offs, except for the χ2 value and the ratio of the χ2 value and degrees of freedom.

Regression analysis showed that gender (b = 1.17, t(1992) = 1.98, *p* = 0.047) and health status (b = 9.10, t(1992) = 10.17, *p* < 0.001) had significant effects on QOLIBRI-KID/ADO total scores. The results indicated that children and adolescents with a chronic health condition had a lower HRQoL compared to those without, while the male gender was associated with a higher HRQoL compared to the female gender. Similar results were found across all scales with statistically significant findings for the influence of health status (*p* < 0.001). Gender did not have a significant effect on the scales, except for the Emotions scale (b = 4.22, t(1992) = 3.95, *p* < 0.001). A second-order regression analysis revealed a significant interaction between age and gender in the total score (b = 3.76, t(1988) = 3.19, *p* = 0.001). Among the scales, a significant interaction was found only for the Emotions (b = 7.61, t(1988) = 3.57, *p* < 0.001) and Physical Problems scales (b = 6.73, t(1992) = 3.08, *p* = 0.002). Further examination of the models for the total score are shown in Table 3. Details on the scales can be found in Appendix A—Table A5 and Table A6. These results suggest that when providing reference values, it is important to separate them by age, gender, and presence of chronic health conditions.

### 3.4. Measurement Invariance across Samples

The baseline and thresholds models did not differ significantly between the TBI sample and the general population sample (ΔCFI < 0.01, ΔRMSEA = 0.001, *p* = 0.064). However, there was a significant difference between the thresholds model and the thresholds and loadings model (ΔCFI = −0.002, ΔRMSEA = −0.002, *p* < 0.001) (Table 4). As such, this implies that the models are not equivalent and that variations in QOLIBRI-KID/ADO scores cannot be attributed to “true” construct differences [55]. However, a closer examination of the differences in thresholds between the two significantly differing models (i.e., thresholds model vs. thresholds and loadings model) in the two samples showed that most of the differences in thresholds were less than 5% (see Appendix A—Figure A1). Figure 2 shows the differences in the factor loadings between the two models. For most of the items, the confidence intervals of the factor loadings overlapped and followed a similar pattern. The exceptions were Concentration, Remembering, Planning (only for the thresholds and loadings model), Appearance, Daily independence, Manage at school, Decision making, Family relationship, Attitudes of others (only for the thresholds and loadings model), Anxiety, Sadness, Anger, Other injuries, Headaches, Pain, and Life changes (only for the thresholds and loadings model). Under these circumstances and considering the cut-off values of ΔCFI and ΔRMSEA, it can be concluded that the construct of HRQoL is largely comparable in both groups.

### 3.5. Reference Values from the General Population Sample

Although our regression analyses showed that chronic health conditions have an impact on HRQoL, we cannot provide separate reference values for individuals from this subgroup due to the small sample size compared to healthy subjects. In Table 5, we present the reference values for the QOLIBRI-KID/ADO total score for the general pediatric population in good health (i.e., without any chronic health conditions, representing an ideal health norm) stratified by gender and age. Due to the small sample size (n = 1), no references could be provided for the group of diverse participants. The interpretation can be performed as follows: assuming a total score of 83 on the QOLIBRI-KID/ADO for a 15-year-old male post-TBI patient, his score falls within the 70% and 85% percentiles compared to a general population sample. Therefore, the patient’s HRQoL is within the average range and is not clinically relevant. In conclusion, the patient’s health and well-being do not appear to be a cause for concern. The scale scores can be treated uniformly. In this instance, it may be beneficial to examine a particular symptom domain, such as Cognition or Emotions, for clinical significance to better narrow down potential areas of concern.

Alternatively, a threshold of one standard deviation below the mean can be utilized to identify clinically significant low HRQoL. For scores falling below this threshold, seeking further diagnosis and treatment is indicated. Participants from the general pediatric population are less inclined to report such a HRQoL. For a male adolescent, critical HRQoL can identified with a cut-off value of 64 for the total score, while for the Cognition scale, it is 61; for the Self scale, it is 60; for the Daily Life and Autonomy scale, it is 68; for the Social Relationships scale, it is 62; for the Emotions scale, it is 50; for the Physical Problems scale, it is 46; and so on. An interactive web application for the reference values tables is available at https://reference-values.shinyapps.io/Tables_Reference_values/ (tab “QOLIBRI-KID/ADO”, last accessed on 4 December 2023).

If these reference values are applied to the TBI sample used in this study, 11.67% of the children and adolescents have a total score below the average (<−1 SD), 76% of the children fall within the average range (±1 SD), and 12.33% of the children are above the average range (>+1 SD).

## 4. Discussion

The aim of this study was to provide reference values for the QOLIBRI-KID/ADO obtained from a German-speaking general population of children and adolescents. For this purpose, we examined its psychometric properties, including reliability and factorial and construct validity. We carried out MI analyses to compare the assessment of HRQoL between the general population and the TBI samples. The QOLIBRI-KID/ADO represents the first TBI-specific pediatric PROM developed to measure HRQoL in children and adolescents post TBI. We adjusted TBI-related content to suit the general population of children and adolescents and found that the QOLIBRI-KID/ADO is a reliable and valid instrument for evaluating HRQoL. MI analyses revealed that HRQoL is assessed similarly in both the general and TBI samples, with minor limitations, enabling fair score comparisons. We present the reference values stratified by age and gender. The analyses suggest that the QOLIBRI-KID/ADO can be applied to the general pediatric population to provide reference values for research and clinical practice, but further research and discussion is needed to address certain issues.

Although DIF analyses revealed significant differences for the age groups, these differences can be considered negligible due to very small effect sizes. Thus, the questionnaire is applicable to children and adolescents, as Steinbuechel et al. [27] found for the TBI version of the QOLIBRI-KID/ADO. They also found a slightly larger effect for the item Daily independence, which was still considered a small effect requiring no further differentiation between children and adolescents. DIF analyses between the samples revealed small significant effects for the items Accomplishment, Appearance, Daily independence, Family relationship, and Other injuries. The small effects in these items could be due to a variety of reasons. A meta-analysis found that children and adolescents with various chronic diseases had a lower body image than healthy peers [56]. Although body image was more negatively distorted for conditions that affect physical appearance (e.g., obesity, scoliosis), less positive body image was also found for almost invisible conditions such as diabetes. Given that TBI is a chronic condition, it is likely that the body image of TBI patients will be lower than that of the general population. Its limited impact on HRQoL may be attributed to the mild TBI of the majority of study participants. They typically experience fewer or less severe symptoms of TBI [57] and, as a result, presumably undergo fewer changes in their appearance. As previous literature has shown [3,4], TBI affects not only the individual, but also their family. General worry was common among families, and they reported interference with daily routines and/or concentration [3], especially when healthcare needs were not covered. Additionally, the severity of TBI was positively associated with limitations in daily routines: the greater the severity, the more families demonstrated a disruption in their daily routines. Interference was found to be correlated with a decrease in PedsQL scores. Although this study only investigated family burden up to one year after TBI, it suggests that these factors should be recognized as early as possible to avoid long-term burden on the family. Therefore, changes in family dynamics may affect the child’s HRQoL years later because the child has been confronted with a serious illness. This, in turn, may impact satisfaction with daily independence and functioning, including accomplishment. Finally, it is possible that children and adolescents in the general population did not suffer any (other) injuries or that they were so minimal that they did not affect HRQoL. Taken together, this may explain the different response behavior between the two samples for these items. However, given that the effects were negligible according to the pseudo-R^2^ cut-off, DIF between the two samples can be considered absent.

The regression analyses revealed similar factors influencing the total HRQoL score as in the previous literature, e.g., in [58,59]. In the current study, total HRQoL decreased with age and this decrease was greater in girls than in boys. There are studies investigating HRQoL that observe an interaction between age and gender [58,60], but the evidence on the influence of age and/or gender seems to be inconsistent. Looking more closely at the significant interactions, the overall finding is a decreasing HRQoL, with a greater decrease for girls than for boys, although this differs for the individual scales. For example, Baumgarten et al. [60] found that in a representative German sample, younger boys experience worse HRQoL in terms of social support and school environment compared to girls. However, these differences become less significant over time. Additionally, girls tend to have lower HRQoL in adulthood on the Physical Well-Being, Psychological Well-Being, and Parents Relation & Autonomy scales, which is consistent with the general trend of lower HRQoL for girls. Ravens-Sieberer et al. [61] found comparable results in a sample of 13 European countries. Bisegger et al. [62] analyzed seven European countries and found no significant effect of age or gender on HRQoL regarding social support and peers. They also found no significant effect of gender on HRQoL related to school environment and no significant difference on HRQoL related to psychological well-being. Therefore, the results for the interaction of age and gender on HRQoL are mixed and it is difficult to draw consistent conclusions, especially because of the different age groups analyzed in the research studies and the different instruments used to measure HRQoL.

Additionally, existing chronic health conditions like allergies or asthma negatively affect HRQoL. Previous research has demonstrated this phenomenon for numerous chronic health conditions, as seen, e.g., in [59,60], and following TBI [11,12,25]. Given these results, a stratification of the reference values by age, gender, and chronic health status seems indicated. We could not provide separated reference values for children and adolescents with chronic health conditions due to the small sample size compared to the sample size of individuals without chronic health conditions. Further research should focus on providing reference values for individuals with chronic health conditions, as chronic conditions become more prevalent in the pediatric population [63,64].

The MI analysis revealed significant findings, indicating that variances in HRQoL between the TBI sample and the general population sample arise from dissimilar compositions or evaluations of the construct [55]. The χ2 difference test analysis revealed a significant result. Here, it is recognized that the χ2 difference test has a high sensitivity to large sample sizes and may identify non-significant equivalence differences with little practical significance [65]. Further analyses were conducted to assess the significance of the variances. The approximated probabilities of selecting a particular response category revealed minimal differences of less than 5% between groups in almost all cases, rendering them negligible. This method and conclusion have previously been implemented in adult general population samples of the QOLIBRI in the United Kingdom [66], the Netherlands [66], and Italy [67]. Moreover, the factor loadings demonstrated a comparable pattern for both groups. After considering the cut-off values of the fit indices for the MI analyses (i.e., ΔCFI and ΔRMSEA) and the results of the DIF analyses, it was determined that the construct was perceived similarly in both groups with minor differences. As a result, different aspects of HRQoL can be assessed and compared between both samples.

Applying the reference values to the TBI sample used in this study, approx. one out of nine children or adolescents had a total HRQoL below the average. In order to better interpret the impact of TBI on HRQoL, it is important to consider the rehabilitation process, because the need for receiving rehabilitation may negatively affect HRQoL [68]. Furthermore, the HRQoL is better associated with the absence or mild TBI [69]. Upon closer examination, it is evident that the participants in this TBI sample primarily experienced a mild TBI approximately 4.51 (SD = 2.78) years prior, with most having achieved a good level of recovery (KOSCHI score of at least 5a). It is plausible that children and adolescents with more severe TBI may have an even more compromised HRQoL, or that a greater number of children falls below the average. Further research is necessary to investigate the effects of severe TBI on HRQoL in children and adolescents.

### 4.1. Strengths and Limitations

This study aims to address the lack of research on disease-specific assessment of HRQoL following TBI. Another notable strength of this study is the validation of the QOLIBRI-KID/ADO in a large general pediatric population sample. This enables the provision of reference values and promotes a better understanding of the limitations, or lack thereof, of HRQoL after TBI in children and adolescents. This can be useful in both research and clinical settings.

However, some limitations should be mentioned. The survey was conducted online through a panel where the parents of the children and adolescents from the general population in Germany were contacted, introducing the possibility of selection bias towards more privileged social groups [70] who are more likely to participate in online studies [71]. It is possible to question the legitimacy of the data obtained by incentivizing participants with a reward after study completion. An attempt was made to mitigate selection bias by using two different research firms on different platforms. However, the extent of bias remains unknown. The agencies involved did not provide information on invitees and non-participants, rendering a drop-out analysis impossible. Moreover, data collection and verification cannot be monitored [72]. The sample of individuals after TBI utilized in calculating the MI analyses was gathered concurrently with questionnaire development, resulting in a higher number of items than the final questionnaire administered to the general population sample. This could potentially affect the findings. Furthermore, the TBI sample may have been biased due to the low response rate, as discussed earlier [27]. Furthermore, it was comparatively small in relation to the general population sample and not all response categories were utilized. To account for the lack of responses in the TBI sample, the two lowest categories (“Not at all” and “Slightly”) were collapsed prior to conducting MI analyses. Collapsing response categories may lead to lower scale reliability and artificially improve model fit [73]. Despite this, response categories were only modified to ensure consistency in the number of response categories used in both samples and enable MI analyses between samples. Further research involving MI analyses using the full five-point scale is therefore recommended.

### 4.2. Clinical Implications

In the case of TBI, it is unlikely that test results on (possibly impaired) abilities will be available prior to traumatic event, making a before-and-after comparison impossible. Therefore, the results are valuable for clinical practice by providing a reference point for assessing HRQoL after TBI. Most importantly, the comparison allows for the identification of below-average HRQoL, which can be targeted for treatment in practice. The study results indicate that boys generally have a higher HRQoL than girls, and that the HRQoL in girls decreases more with age. Additionally, this study found that health status, especially the presence of at least one chronic health condition, has a negative impact on HRQoL. These findings suggest that health policies should focus more on improving the HRQoL of adolescent girls and addressing chronic diseases in both childhood and adolescence.

The objective of healthcare and rehabilitation is to restore an individual’s full health or enable them to achieve the highest possible HRQoL. TBI can result in various impairments, e.g., cognitive, emotional, and behavioral. By utilizing the reference values, healthcare professionals can improve TBI treatment by more accurately assessing areas of impairment in direct comparison to those without TBI. These reference values indicate a desirable health condition and provide feedback to healthcare professionals regarding areas that require support due to impairment. This allows for more individualized treatment for patients following TBI. Since this is a disease-specific tool, it is not applicable to other chronic diseases as it does not address the specific symptoms of those conditions.

## 5. Conclusions

The QOLIBRI-KID/ADO is a valid tool for evaluating disease-specific HRQoL in children and adolescents after TBI. An adapted version of the instrument is applicable for the pediatric population, allowing for meaningful comparisons between children and adolescents with and without TBI and serving as a reference for interpretation of QOLIBRI-KID/ADO scores. The use of reference values in clinical practice can improve the assessment of disease-specific HRQoL and the evaluation of children and adolescents with TBI. Future research should focus on developing reference values for the general German pediatric population affected by chronic health conditions.

## Figures and Tables

**Figure 1 jpm-14-00336-f001:**
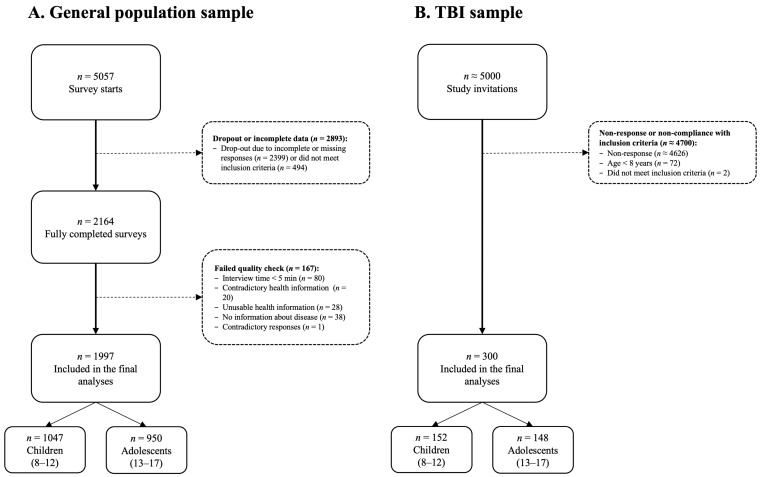
Flow chart of sample composition: (**A**) general population sample and (**B**) TBI sample.

**Figure 2 jpm-14-00336-f002:**
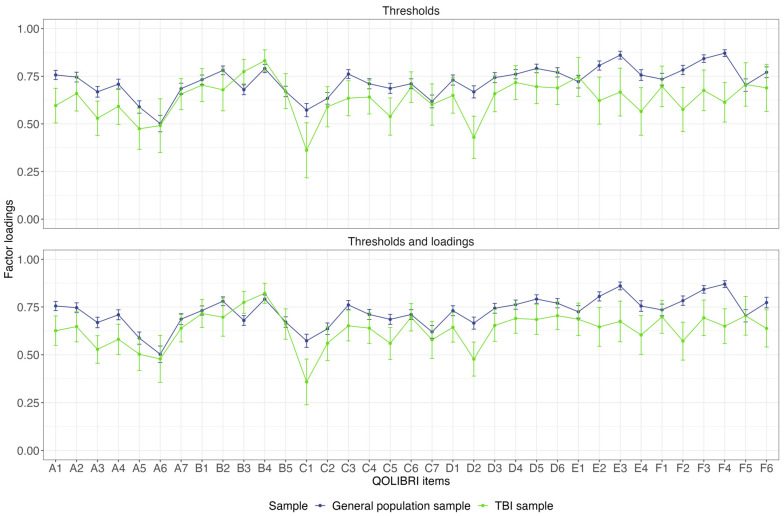
Comparison of standardized factor loadings between models with increasing constraints (i.e., thresholds model vs. thresholds and loadings model) between general population and TBI sample.

**Table 1 jpm-14-00336-t001:** Sociodemographic and health-related data of the general population sample.

		Children (*n* = 1047)	Adolescents (*n* = 950)	Total (*n* = 1997)
Age (years)	Mean (SD)	10.0 (1.42)	15.0 (1.39)	12.4 (2.85)
	Median [min, max]	10.0 [8.00, 12.0]	15.0 [13.0, 17.0]	12.0 [8.00, 17.0]
Gender	Female	523 (50.0%)	484 (50.9%)	1007 (50.4%)
	Male	524 (50.0%)	465 (48.9%)	989 (49.5%)
	Diverse	0 (0%)	1 (0.1%)	1 (0.1%)
Education	None	0 (0%)	6 (0.6%)	6 (0.3%)
	Not identified *	15 (1.4%)	15 (1.6%)	30 (1.5%)
	Primary school	556 (53.1%)	0 (0%)	556 (27.8%)
	Special school	47 (4.5%)	36 (3.8%)	83 (4.2%)
	Secondary school	46 (4.4%)	73 (7.7%)	119 (6.0%)
	Secondary school/middle school	190 (18.1%)	343 (36.1%)	533 (26.7%)
	Vocational school	0 (0%)	78 (8.2%)	78 (3.9%)
	Preparatory high school	193 (18.4%)	399 (42.0%)	592 (29.6%)
Integration assistance at school	Yes	145 (13.8%)	125 (13.2%)	270 (13.5%)
	No	902 (86.2%)	819 (86.2%)	1721 (86.2%)
	Missing	0 (0%)	6 (0.6%)	6 (0.3%)
Number of chronic health conditions	One and more	122 (11.7%)	127 (13.4%)	249 (12.5%)
	None	925 (88.3%)	823 (86.6%)	1748 (87.5%)

Note. *: Due to implausible data (8–12 years and vocational school and 13–17 years and primary school, which is very unlikely in the German school system), the category “not identifiable” was added. SD = standard deviation, *n* = sample size.

**Table 2 jpm-14-00336-t002:** Psychometric properties of the QOLIBRI-KID/ADO in the general population sample.

Scale	Item	Cronbach’s α ^a^	McDonald’s ω	Alpha if Item Omitted ^a^	Item–Total Correlations ^a^	CITC
Cognition		0.80	0.84			
	Concentration			0.76	0.75	0.63
	Talking to others			0.77	0.69	0.55
	Remembering			0.76	0.71	0.58
	Planning			0.77	0.7	0.57
	Decision between two			0.79	0.62	0.46
	Orientation			0.80	0.54	0.36
	Thinking speed			0.76	0.71	0.58
Self		0.80	0.83			
	Energy			0.77	0.73	0.55
	Accomplishment			0.76	0.75	0.59
	Appearance			0.77	0.74	0.57
	Self-esteem			0.74	0.79	0.65
	Future			0.77	0.72	0.55
Daily Life and Autonomy		0.80	0.83			
	Daily independence			0.79	0.63	0.47
	Getting out and about			0.77	0.72	0.59
	Manage at school			0.78	0.67	0.53
	Social activities			0.77	0.72	0.60
	Decision making			0.77	0.71	0.57
	Support from others			0.78	0.66	0.51
	Ability to move *			0.79	0.63	0.47
Social Relationships		0.84	0.86			
	Open up to others			0.81	0.73	0.59
	Family relationship			0.82	0.69	0.54
	Relationship with friends			0.81	0.76	0.63
	Friendships			0.80	0.77	0.64
	Attitudes of others			0.80	0.79	0.68
	Demands from others			0.82	0.72	0.58
Emotions		0.82	0.84			
	Loneliness			0.81	0.76	0.57
	Anxiety			0.76	0.84	0.69
	Sadness			0.73	0.87	0.75
	Anger			0.81	0.77	0.59
Physical Problems		0.86	0.90			
	Clumsiness			0.86	0.68	0.54
	Other injuries *			0.83	0.81	0.71
	Headaches			0.82	0.83	0.74
	Pain			0.82	0.86	0.78
	Seeing/Hearing			0.85	0.73	0.60
	Life changes *			0.85	0.71	0.58
Total score		0.94	0.95			

Note. *: Reworded items. ^a^ Standardized alpha coefficients are reported. CITC: corrected item–total correlations.

**Table 3 jpm-14-00336-t003:** Results of regression analysis for the QOLIBRI-KID/ADO total score.

Model	Variable	Reference Category	*b*	S.E.	*t*	*p*
Model without interactions	Intercept	-	64.52	0.94	68.68	**<0.001**
	Age group	Children (8–12 years)	−0.15	0.59	−0.25	0.801
	Gender	Female	1.17	0.59	1.98	**0.047**
	Health status	At least one chronic health condition	9.10	0.89	10.17	**<0.001**
Model with interactions	Intercept	-	65.32	1.49	43.76	**<0.001**
	Age group	Children (8–12 years)	−4.19	1.77	−2.37	**0.018**
	Gender	Female	1.72	1.77	0.97	0.333
	Health status	At least one chronic health condition	9.20	1.56	5.91	**<0.001**
	Age group * Gender	Children * Female	3.76	1.18	3.19	**0.001**
	Age group * Health status	Children * At least one chronic health condition	2.49	1.78	1.40	0.163
	Gender * Health status	Female * At least one chronic health condition	−2.66	1.78	−1.49	0.136

Note. b: non-standardized regression coefficient; S.E.: standard error; t: t-value; *p*: *p*-value; values in **bold** are significant at 5%; *: interaction between the variables.

**Table 4 jpm-14-00336-t004:** Results of measurement invariance (MI) analyses.

Samples	Constraints	χ2 (df)	*p*	χ2/df	CFI	TLI	RMSEA [90% CI]	SRMR	χ2 (df)	Δ χ2	Δ df	*p*
General population sample vs. TBI sample	baseline	4006.65 (1090)	**<0.001**	3.77	**0.96**	**0.95**	**0.049 [0.047, 0.050]**	**0.054**	4476.6 (1090)	-	-	-
	thresholds	4042.66 (1125)	**<0.001**	3.59	**0.96**	**0.95**	**0.048 [0.046, 0.049]**	**0.054**	4498.7 (1125)	48.522	35	0.064
	thresholds and loadings	3973.81 (1154)	**<0.001**	3.44	**0.96**	**0.96**	**0.046 [0.045, 0.048]**	**0.054**	4620.0 (1154)	75.773	29	<0.001

Note. χ2: scaled chi-square statistics; df: scaled degrees of freedom; *p*: *p*-value; χ2/df: scaled ratio (cut-off: ≤2); CFI: scaled comparative fit index (cut-off: >0.90); TLI: scaled Tucker–Lewis index (cut-off: >0.95); RMSEA [90% CI]: scaled root mean square error of approximation with 90% confidence interval (cut-off: <0.06); SRMR: scaled standardized root mean square residual (cut-off: <0.08). Values in **bold** indicate at least a satisfactory/mediocre model fit according to the respective cut-offs and/or are within acceptable range.

**Table 5 jpm-14-00336-t005:** Reference values for the QOLIBRI-KID/ADO for the general population without chronic health conditions.

				Low Quality of Life		−1 SD			Md			+1 SD		High Quality of Life
Gender	Age	Scale	*n*	2.50%	5%	16%	30%	40%	50%	60%	70%	85%	95%	97.50%
Male	8–12	Total Score	462	50	55	62	68	71	73	76	80	86	95	100
		Cognition		46	50	64	68	71	75	79	82	89	100	100
		Self		45	50	60	70	70	75	80	85	95	100	100
		Daily Life and Autonomy		50	57	68	71	75	79	82	86	93	100	100
		Social Relationships		50	54	67	71	75	79	83	88	96	100	100
		Emotions		12	25	44	52	62	69	75	81	94	100	100
		Physical Problems		12	17	38	50	58	67	75	79	92	100	100
	13–17	Total Score	401	46	52	64	70	74	76	79	81	89	97	100
		Cognition		43	50	61	71	75	79	82	86	93	100	100
		Self		40	50	60	70	70	75	75	80	90	100	100
		Daily Life and Autonomy		46	50	68	75	79	82	86	89	96	100	100
		Social Relationships		50	50	62	71	75	79	83	88	96	100	100
		Emotions		25	31	50	62	69	75	81	88	94	100	100
		Physical Problems		17	25	46	58	67	75	79	88	96	100	100
Female	8–12	Total Score	463	48	54	62	68	71	75	78	82	87	96	100
		Cognition		48	54	64	70	75	75	82	82	89	100	100
		Self		40	50	65	70	75	75	80	85	95	100	100
		Daily Life and Autonomy		50	54	68	75	79	82	86	89	93	100	100
		Social Relationships		46	50	67	71	75	79	83	88	92	100	100
		Emotions		12	19	38	56	62	69	75	81	94	100	100
		Physical Problems		12	21	37	54	62	71	75	83	92	100	100
	13–17	Total Score	422	47	49	61	66	69	73	76	81	88	95	97
		Cognition		46	50	61	68	71	75	79	82	89	96	100
		Self		40	45	55	65	70	75	78	85	90	95	100
		Daily Life and Autonomy		46	50	68	72	75	82	86	89	96	100	100
		Social Relationships		42	50	67	71	75	79	82	88	92	100	100
		Emotions		12	19	38	50	62	69	75	81	94	100	100
		Physical Problems		17	21	38	50	58	67	75	83	92	100	100
		Total Score	1748	47	53	62	68	71	74	77	81	88	96	100

Note. 50% percentiles represent 50% of the distribution, corresponding to both the median (Md) and the mean; SD: standard deviation; values from −1 standard deviation (16% rounded up to the next integer) to +1 standard deviation (85% rounded up to the next integer) are within the normal range; values below 16% indicate low quality of life and values above 85% indicate high quality of life.

## Data Availability

The data presented in this study are available upon request from the corresponding author. Data are not publicly available for privacy reasons. R scripts are available from GitHub https://github.com/mzeldovich/Project-Reference-values (last accessed on 4 December 2023). The R-Shiny application with reference values is available at https://reference-values.shinyapps.io/Tables_Reference_values/ (tab QOLIBRI-KID/ADO; last accessed on 4 December 2023).

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
