# Peer review of "Reference Values for the German Version of the Quality of Life after Brain Injury in Children and Adolescents (QOLIBRI-KID/ADO) from a General Population Sample"

_jpm, 2024, doi:10.3390/jpm14040336_

Round 1

Reviewer 1 Report

Comments and Suggestions for Authors

Summary:

This research aims to establish reference values for the QOLIBRI-KID/ADO in German-speaking children and adolescents. The results demonstrate that the instrument is reliable and valid for assessing health-related quality of life (HRQoL) in this population. Analyses reveal that HRQoL is assessed similarly in samples from the general population and traumatic brain injury (TBI), allowing for fair comparisons between groups. While the study is enlightening, there are areas, as pointed out in the review, where clarity and structure could be enhanced to improve the paper's coherence and interpretability.

Comments:

Introduction section

1. The transition from discussing generic HRQoL measures to introducing the QOLIBRI-KID/ADO questionnaire could be smoother. It would be interesting to demonstrate the gap more clearly in available measures for assessing HRQoL in pediatric TBI and how the QOLIBRI-KID/ADO fills that gap.

2. Although you briefly mention the development process of the QOLIBRI-KID/ADO, I suggest providing a bit more detail about its characteristics and how it differs from generic HRQoL measures.

Methods section

1. The sample sizes of each group were quite different. Additionally, while the general population sample was recruited through marketing agencies, the TBI sample was collected in hospitals in Germany over a period of several years. Do you believe these differences could introduce bias into the results?

Results section

1. How can the results of this study be applied in clinical practice or in the formulation of health policies? Are there any important practical implications resulting from these results?

Discussion section

1. How can healthcare professionals utilize the findings of your study to enhance the care and treatment of children and adolescents with TBI or other chronic conditions?

Reviewer 2 Report

Comments and Suggestions for Authors

1. In Table A3, why the P values (=.001) of the items “Decision between two” and “Attitudes of others” were not in bold? In Table A4, why the P values (=.009) of the items “Getting out and about” were not in bold?

2. In section 3.3, since the authors used the unstandardized regression coefficients, the symbol “β” should be changed into “b” to avoid being misunderstood.

3. In lines 341-343, the authors said “However, a closer examination of the differences in thresholds between the two significant models in the two samples showed that most of the differences were less than 5%”. Did it mean that “less than 5% number of items”?

4. Besides, in Figure A2, what did the horizontal axis mean? the positive value meant which sample (?) was higher than the other sample (?) in the horizontal value? That is, please also denote what the positive value of the horizontal axis means.

5. In Table 5, the title “…for the general population” would be misunderstood since in line 366, those with chronic health conditions were not considered in Table 5. The title should be corrected for this point to avoid misunderstanding. Besides, the topic of the manuscript “…from a general population sample” should also be cautiously used.

6. In line 399, “We present reference values stratified by age and gender”. Please also denote it again in the websie “https://reference-values.shinyapps.io/Tables_Reference_values/”

  7. in the abstract, the authors said, “Reference values from a comparable general population are essential for comprehending the impact of TBI on health and well-being.” How about adding to the report the percentages of the three zones (below 1 SD, between +-1 SD, and above 1 SD) for the TBI sample based on the Norm ( general population, N=1748) in the manuscript? And some impacts or discoveries can be written out.
